# The Effect of Titanium Dioxide Surface Modification on the Dispersion, Morphology, and Mechanical Properties of Recycled PP/PET/TiO_2_ PBNANOs

**DOI:** 10.3390/polym11101692

**Published:** 2019-10-16

**Authors:** Eider Matxinandiarena, Agurtzane Múgica, Manuela Zubitur, Cristina Yus, Víctor Sebastián, Silvia Irusta, Alfonso David Loaeza, Orlando Santana, Maria Lluisa Maspoch, Cristian Puig, Alejandro J. Müller

**Affiliations:** 1POLYMAT and Polymer Science and Technology Department, Faculty of Chemistry, University of the Basque Country UPV/EHU, Paseo Manuel de Lardizabal 3, 20018 Donostia-San Sebastián, Spain; eider.matxinandiarena@ehu.eus (E.M.); agurtzane.mugica@ehu.eus (A.M.); 2Chemical and Environmental Engineering Department, Polytechnic School, University of the Basque Country UPV/EHU, 20018 Donostia-San Sebastián, Spain; manuela.zubitur@ehu.eus; 3Department of Chemical and Environmental Engineering, Nanoscience Institute of Aragon University of Zaragoza and, Aragón Materials Science Institute, ICMA, CSIC, Pedro Cerbuna 12, 50009 Zaragoza, Spain; cyargon@gmail.com (C.Y.); victorse@unizar.es (V.S.); sirusta@unizar.es (S.I.); 4Networking Research Center CIBER-BBN, 28029 Madrid, Spain; 5Centre Català del Plàstic (CCP), Universitat Politècnica de Catalunya Barcelona Tech (EEBE-UPC), C/Colom, 114, 08222 Terrassa, Spain; alfonso.david.loaeza@upc.edu (A.D.L.); orlando.santana@upc.edu (O.S.); maria.lluisa.maspoch@upc.edu (M.L.M.); 6Grupo de Polímeros USB, Departamento de Ciencias de los Materiales, Universidad Simón Bolívar, Apartado 89000, Caracas 1080A, Venezuela; cpuig@usb.ve; 7IKERBASQUE, Basque Foundation for Science, 48013 Bilbao, Spain

**Keywords:** blend, nanoparticles, titanium dioxide, morphology, PBNANO

## Abstract

Titanium dioxide (TiO_2_) nanoparticles have recently appeared in PET waste because of the introduction of opaque PET bottles. We prepare polymer blend nanocomposites (PBNANOs) by adding hydrophilic (hphi), hydrophobic (hpho), and hydrophobically modified (hphoM) titanium dioxide (TiO_2_) nanoparticles to 80rPP/20rPET recycled blends. Contact angle measurements show that the degree of hydrophilicity of TiO_2_ decreases in the order hphi > hpho > hphoM. A reduction of rPET droplet size occurs with the addition of TiO_2_ nanoparticles. The hydrophilic/hydrophobic balance controls the nanoparticles location. Transmission electron microscopy (TEM_ shows that hphi TiO_2_ preferentially locates inside the PET droplets and hpho at both the interface and PP matrix. HphoM also locates within the PP matrix and at the interface, but large loadings (12%) can completely cover the surfaces of the droplets forming a physical barrier that avoids coalescence, leading to the formation of smaller droplets. A good correlation is found between the crystallization rate of PET (determined by DSC) and nanoparticles location, where hphi TiO_2_ induces the highest PET crystallization rate. PET lamellar morphology (revealed by TEM) is also dependent on particle location. The mechanical behavior improves in the elastic regime with TiO_2_ addition, but the plastic deformation of the material is limited and strongly depends on the type of TiO_2_ employed.

## 1. Introduction

Plastic consumption in packaging and many other applications is increasingly growing. In the year 2017, 348 million tones of plastics were produced all around the world, from which 64.4 million tones (18.5%) were produced in Europe. As a consequence, a large amount of plastic waste is generated, 27.1 million tones of those in the European Union [1]. Although recycling is the first option for plastic packaging waste (40.8% recycling, 38.8% energy recovery, 20.4% landfill), these figures show that the recycling rates are still low and that more efficient reprocessing techniques are needed to make the recycling process more sustainable [1].

Recycling of waste plastic has become a worldwide trend, because of several benefits like saving manufacturing resources, lowering energy consumption, and minimizing the impact of plastics on the environment [2]. Polyethylene terephthalate (PET) is one of the most widely used polymers in packaging, because of its good tensile strength, barrier properties, chemical resistance, elasticity, and the wide temperature range that it can support [3].

Many methods have been developed to recycle PET waste. Mechanical recycling (blending it with other polymers) is one of the most popular, because it is easy, economically affordable, and materials with relatively good mechanical properties can be obtained [2]. Also, it is very common to find more than one type of plastic in packaging products, such as water or milk bottles with polyethylene terephthalate (PET) bodies and polypropylene (PP) or polyethylene (PE) caps [4]. This implies removing the caps from the bottles before recycling or recycling PET/PP or PET/PE immiscible blends. Opaque PET bottles that have been introduced in the past few years in the market (for milk bottles, oil bottles, etc.) contain TiO_2_ in quantities that vary depending on the application (but they can be as high as 15%). The TiO_2_ is used because it protects the fluids in the bottle from UV radiation [5].

Immiscible polymer blends are characterized by high interfacial tension and poor adhesion between the phases [6,7,8]. Compatibilization methods to reduce the interfacial tension and enhance adhesion between phases are usually required. The addition of nanofillers to polymer blends is an alternative way to compatibilization, as the filler can act as a Pickering emulsion promoter [9,10,11]. This means that if the nanofiller can be located at the interface between the two immiscible phases, then a reduction of the surface tension can be achieved with a concomitant reduction in particle sizes for blends with sea-island morphologies (typically with more than 60% of one of the components in the blends). Furthermore, the new inter-phase constituted by the surrounding nanofiller shell can act as a physical barrier that prevents coalescence during processing or post-processing procedures. The blends with added nanofillers have been denominated PBNANOs, i.e., polymer blend nanocomposites [12]. The changes in morphology caused by the addition of nanofillers, such as TiO_2_, can produce significant effects on the mechanical, thermal, barrier, and electrical properties of PBNANOs [13,14,15].

One of the key factors in PBNANOs morphology and therefore in their properties is the dispersion and specific location of the nanofillers. TiO_2_ nanoparticles can be located in one of the phases, at the interface between the matrix and the dispersed phase, or in both places at the same time. The nanofiller location will be determined by the affinity of the nanoparticles to each phase [3,16].

Some works have already been reported for PET/polyolefin/TiO_2_ polymer blends. Sangroniz et al. studied the relationship between morphology, thermal properties, and rheology in PET/LDPE and PET/LDPE/TiO_2_ PBNANOs rich in PET. They reported a reduction of droplet size with the addition of TiO_2_ because of its location at the interphase, and a moderate nucleation effect [3].

Several works have been published dealing with the preferential location of nanofillers in immiscible blends [17,18,19,20,21]. Nanofillers are usually functionalized at their surface so that they can have more affinity to one phase. Elias et al. studied the location of hydrophilic and hydrophobic silica in a polypropylene/polystyrene blend. The hydrophilic silica was mostly located within the PS phase, whereas the hydrophobic one was located within the PP phase. They attributed the preferential location to the lower interfacial tension between the nanoparticles and the specific phase where it was located [22]. Majesté et al. reported the confinement of hydrophilic silica in the EVA phase for PP/EVA immiscible blends, whereas hydrophobic nanosilica treated with trimethoxyoctylsilane locates at the PP/EVA interphase [23].

However, other authors reported that surface tension is not always the key factor responsible for the preferential location of nanofillers in a blend of two chemically different polymers. The flexibility of polymer chains also has to be considered, as the entropy penalty may be decisive in the absorption of polymers on the surfaces of many nanofillers [24].

In addition to the above mentioned thermodynamic factors, the blending sequence could also be investigated to determine the preferential location of nanofillers. There are two main methods to incorporate nanofillers into blends: mixing the three components at the same time or, blending one polymer with the nanofiller and adding the second polymer in another extrusion process. Some investigations have shown that the blending sequence does not affect the preferential location of the nanofillers, as they will stay or migrate to the preferred phase [9,25].

Fu et al. studied the kinetic-controlled compatibilization of PP/PS blends with nanosilica. According to them, the mixing time affects the dispersed droplet size, as shorter times lead to a reduction of the dispersed PS droplet size [26].

Many reports discuss the mechanical properties of filled polymer systems. Fillers have been used to modify many properties of polymers, such as strength, toughness, processability, dimensional stability, or lubrication. The interaction with the polymer matrix and the filler concentration and particle size are important factors to improve the mechanical properties. Modification of the surface of fillers is becoming more important because of its improvement on adhesion, and hence, on the stress transfer between polymer and filler which leads to an increase in the dispersion degree [27]. For instance, organically modified montmorillonite is widely used as a filler in many systems. Aubry et al. studied the effect of adding a modified montmorillonite to an 80% PE/20% PA blend. Because of the location of nanoclays at the interface, break up was promoted and coalescence inhibited stabilizing the system. Mechanical properties such as stress and elongation at break were improved, as energy dissipation was favored, making these organo-nanoclays good compatibilizers [28,29].

In the present work, TiO_2_ nanoparticles are added to recycled PET (rPET) and recycled PP (rPP) to prepare 80rPP/20rPET/TiO_2_ PBNANOs. Three different types of nanoparticles are used: hydrophilic TiO_2_ (hphi), hydrophobic TiO_2_ (hpho), and hydrophobically modified TiO_2_ (hphoM) nanoparticles by a sol-gel reaction. We are interested in studying the preferential location of the nanoparticles and their effect on morphology, crystallization kinetics, and mechanical properties of the PBNANOs.

## 2. Materials and Methods

### 2.1. Material Part

#### 2.1.1. Materials

The materials used in this work are recycled PET flakes (rPET) and recycled PP pellets (rPP) provided by *Suez* (a waste treatment French company). It is important to note that the recycled PET material already contains 2% of a commercial hydrophobic TiO_2_. Three different titanium dioxide nanoparticles were used: (a) A hydrophilic titanium dioxide (hphi) 99%, provided by io-li-tec nanomaterials (a German company that sells ionic liquids and nanoparticles in Heilbronn, Germany) with rutile crystalline form; (b) a hydrophobic titanium dioxide (hpho) 92%, coated with silicone oil from US Research Nanomaterials, Inc. (Houston, TX, USA), with rutile crystalline form; and (c) hydrophilic titanium dioxide (AEROXIDE TiO_2_ P25 99.5% from Evonik Industries, Essen, Germany) modified by a sol-gel reaction with octadecyltrimethoxysilane (97% Apollo Scientific, Manchester, UK) to turn it hydrophobic, this is denoted hydrophobically modified TiO_2_ (hphoM).

#### 2.1.2. TiO_2_ Nanoparticles Modification

The hydrophobization process of TiO_2_ nanoparticles was carried out following the procedure reported by Pazokifard et al. [30]. Briefly, 1 g of particles were dispersed in 25 mL ethanol and then a 1 mmol octadecyltrimethoxysilane (TMOS) solution in ethanol was added. The pH was adjusted to 12 using NH_4_OH. The mixture was kept for 18 h under stirring. Then the particles were washed by centrifugation (8000 rpm, 10 min) and dried at 60 °C for 24 h (Appendix A).

#### 2.1.3. PBNANOs Preparation

Materials were dried for 48 h at 80 °C under vacuum before extrusion. 80rPP/20rPET/TiO_2_ (*w*/*w*) PBNANOs were prepared in a Collins ZK 25T co-rotating twin-screw extruder with a temperature profile of 215–270 °C and 40 rpm. Extrudates were cooled in a water bath and cut into pellets by a pelletizer. First, rPP/10%TiO_2_ and rPP/20%TiO_2_ masterbatches were prepared. Then, adequate quantities of the masterbatch were used to increase the amount of titanium dioxide by 1, 3, 5, 7.5, and 10% with respect to the total amount of material, so that polymer blends with the same polymer ratio of 80rPP/20rPET could be prepared. As the 80rPP/20rPET blend has approximately 2% TiO_2_ nanoparticles (as a residue in the rPET employed), the final concentrations of TiO_2_ are 3, 5, 7, 9.5, and 12% TiO_2_. All PBNANOs prepared are listed in Table 1, it should be noted that three different types of titanium dioxide nanoparticles (as described above) were used to prepare each of the compositions.

Samples for morphological and mechanical characterization were prepared by compression molding using a P200E Collin (Ebersberg, Germany) hot press at 270 °C and under 150 bar for 3 min, and then they were quenched in a water bath. Square plates of 100 × 100 mm and 1 mm in thickness were prepared. Specimens of 1 mm were obtained for SEM observation.

### 2.2. Characterization Part

#### 2.2.1. Nanoparticles Characterization

Water contact angle measurements were carried out to determine the hydrophobicity of the TiO_2_ NPs in a Dataphysics OCA-Series equipment (Dataphysics Instruments GmbH, Filderstadt, Germany) at room temperature. Particle films were deposited on a slide, and the contact angles were calculated from water drop images. The hydrodynamic nanoparticle size and ζ-potential in water were determined by dynamic light scattering (DLS) using a Brookhaven Instruments 90Plus (SRC, Ontario, Canada). The presence of organic compounds on NPs surface was confirmed by Fourier-transform infrared (FTIR) spectra obtained with a Vertex 70 from Bruker, equipped with an ATR Golden Gate accessory. Hydrophobic and hydrophobically modified nanoparticles were characterized with a thermogravimetric analyzer (TGA) from Mettler Toledo (L'Hospitalet de Llobregat, Barcelona) (TGA/SDTA 851e) to measure the organic content. The TGA experiments were performed in an air atmosphere at 10 °C/min in a temperature range from 30 to 800 °C.

#### 2.2.2. Morphological Analysis (SEM, TEM)

The morphology of the blends and nanoparticles was first observed by SEM using a Hitachi S-2700 (Rocklin, CA, USA) electron microscope under high vacuum and with accelerating voltage of 20 kV, at different magnifications. Before observation, the samples were cryogenically fractured in liquid N_2_, and the fracture surfaces were coated with gold in a Bio-Rad SC500 (Lewes, United Kingdom) sputter coater before being exposed to the electron beam. The diameter of rPP droplets was measured on around 100 particles. Number (*d_n_*) and volume (*d_v_*) average diameters and particle size polydispersity (*D*) were calculated using the following Equations (1) [31]:(1)$$d_{n}=\frac{\sum n_{i}d_{i}^{2}}{\sum n_{i}d_{i}}\;d_{v}=\frac{\sum n_{i}d_{i}^{4}}{\sum n_{i}d_{i}^{3}}\;D=\frac{d_{v}}{d_{n}}$$
where *n_i_* is the number of particles of diameter *d_i_*.

The location of TiO_2_ nanoparticles was determined by transmission electron microscopy (TEM) analysis. The samples were first cut at room temperature with a diamond knife on a Leica EMFC 6 ultramicrotome device. The ultra-thin sections of 90-nm thick were mounted on 200 mesh copper grids. Finally, they were examined using a TECNAI G2 20 TWIN TEM equipped with LaB6 filament operating at an accelerating voltage of 120 kV (ThermoFisher Scientific, Waltham, MA, USA). To image the lamellar morphology, a RuO_4_ solution was employed for staining. Thin strips of samples were immersed in this solution for 16 h. Then, ultra-thin sections were cut at room temperature and were analyzed as mentioned above.

#### 2.2.3. Thermogravimetric Analysis (TGA)

The TiO_2_ content in the blend 80rPP/20rPET was determined in a Q500 TA Instruments (New Castle, DE, USA) TGA analyzer under oxidative atmosphere. The sample mass was approximately 5 mg. Before measurement, the samples were dried overnight. A heating run was carried out from 40 to 700 °C at 10 °C min^−1^. The unburnt residue left at 700 °C was measured to determine the final TiO_2_ content. TGA measurements were also performed to check the final content of TiO_2_ nanoparticles in all the prepared blends.

#### 2.2.4. Thermal Analysis (DSC)

The thermal properties of the samples were studied by differential scanning calorimetry (DSC) using a Perkin Elmer (Hopkinton, MA, USA) DSC Pyris 1 equipped with a refrigerated cooling system (Intracooler 2P), under ultra-high purity nitrogen atmosphere. The instrument was previously calibrated with indium and tin standards. Before the measurements, all samples were dried overnight at 80 °C under vacuum. Aluminum pans were used to encapsulate samples of approximately 5 mg. The different thermal protocols used to study the crystallization behavior of the samples are described below.

##### Non-Isothermal DSC Experiments

All samples were first measured by standard non-isothermal DSC experiments. The thermal program applied was the following: an initial heating run from 25 to 270 °C at 20 °C min^−1^ and 3 min at 270 °C to erase the thermal history, then a cooling scan down to −20 °C at 20 °C min^−1^, 1 min of stabilization at −20 °C, and finally, a second heating run to 270 °C at 20 °C min^−1^.

##### Isothermal DSC Experiments

Isothermal measurements were performed using the procedure recommended by Lorenzo et al. [32]. The thermal program applied was the following: erasure of thermal history by heating the samples to 270 °C for 3 min, rapid cooling to a chosen isothermal crystallization temperature (*T_c_*) at a controlled rate of 60 °C min^−1^, isothermal crystallization (measurements of heat flow versus time) at *T_c_* until saturation (typically, the peak time × 3), and final heating run to 270 °C at 20 °C min^−1^. The minimum isothermal crystallization temperature (*T_c_*) employed was the lowest temperature, which did not show any melting enthalpy during immediate subsequent heating.

#### 2.2.5. Mechanical Behavior

The mechanical behavior was studied by tensile tests, according to ISO-527-2 standard, on PBNANOs without and with 10% *w*/*w* of additional TiO_2_ (hydrophobic (hpho), hydrophobically modified (hphoM) and hydrophilic (hphi)). A GALDABINI 2500 (Cardano al Campo, Italy) universal testing machine equipped with a MINTRON OS-65D (New Tapei, Taiwan, China) video extensometer and a 1 kN load cell was used. The specimens were stamped from the compression molded plates according to the type 1 BA dumbbell geometry. The tests were carried out at 22 ± 1 °C and a crosshead speed of 5 mm.min^−1^. The engineering stress–strain curves for every sample were recorded, and elastic modulus (E), yield strength (σ_y_), yield strain (ɛ_y_), and strain at break (ε_b_) were determined. The average values and corresponding standard deviation were determined from 10 valid tests.

After testing, the fractured surfaces of all samples were inspected by scanning electron microscopy (SEM) (JEOL, JSM-7001F, Tokyo, Japan). Experiments were performed under vacuum with an accelerating voltage of 2 kV. Samples were previously coated with platinum vapor.

## 3. Results

### 3.1. Nanoparticles Characterization

SEM and TEM images of the three different types of nanoparticles added to the blends (Figure 1) show that the hydrophilic nanoparticles have larger diameters than the hydrophobic nanoparticles.

However, the commercial hydrophobic NPs present a high degree of agglomeration that was confirmed by DLS analysis, resulting in a hydrodynamic diameter of 541 ± 13 nm (Table 2) and a low ζ-potential (−9.1 ± 1.9 mV). On the other hand, the dimensions of hydrophobically modified nanoparticles were increased by about 42%, whereas the ζ-potential decreased to −31 mV with respect to the pristine P25 nanoparticles. These results point to a successful functionalization process (Table 2). The presence of the octadecyltrimethoxysilane (TMOS) around the hydrophobically modified nanoparticles can be clearly observed in TEM images using a negative staining agent (Figure 1).

The degree of hydrophilicity was determined by contact angle measurements. A water drop was immediately absorbed by the film with a water contact angle (WCA) of almost 0° for the un-modified particles, while the hydrophilic ones showed a WCA of around 36°. This behavior was expected because of the presence of a large number of hydroxyl groups on the surface of the particles [33]. After modification with TMOS a completely different behavior was observed, the water contact angle increased to 144° because of the replacement of OH groups by hydrophobic molecules. The commercial hydrophobic nanoparticles showed a WCA lower than the modified ones, but they can still be considered as hydrophobic nanoparticles [34].

The presence of the organic compounds on the surface of the particles was also confirmed by FTIR. Figure 2a shows the spectra of non-modified and modified particles together with TMOS spectrum. The band at 1100 cm^−1^, characteristic of Si-O-CH_3_ stretching, indicates the presence of the methyl-silane groups. Bands at 1200 and 1467 cm^−1^ could be assigned to CH_3_ rocking and CH_2_ stretching respectively, and bands in the range 3000–2780 cm^−1^ are due to C–H bonds stretching. It is important to notice the presence of a wide band around 3250 cm^−1^ that would be related to O–H stretching of silanol groups derived from the partial hydrolysis of Si–O–CH_3_ [35]. On the other hand, the commercial hydrophobic particles spectrum shows only the presence of organic groups (1100–1000 cm^−1^), but no bands related to O–H groups could be observed (Figure 2a).

The load of organic compound in the hydrophobic (hpho) and functionalized nanoparticles (hphoM), obtained from TGA results, was 1.6 ± 2% and 16 ± 3%% respectively (Figure 2b).

### 3.2. Blends Morphology

All PBNANOs prepared in this work exhibit a sea-island or droplet-matrix morphology, as expected for immiscible blends, with rPP as the matrix (80%) and rPET (20%) as the dispersed droplets. Figure 3 shows a SEM micrograph of the extruded 80rPP/20rPET/2%TiO_2_ blend. TiO_2_ nanoparticles were not added in the preparation of this blend, as the recycled PET (rPET) already included 2% TiO_2_ nanoparticles. The TiO_2_ nanoparticles content was determined by TGA. A lack of adhesion between rPP matrix and rPET droplets can be observed, indicated by the presence of holes, where particles were detached. Both in the holes and on some of the particle surfaces, very small white particles can be observed, which are the TiO_2_ nanoparticles [36]. In some cases, TiO_2_ aggregates can also be observed. The location of TiO_2_ nanoparticles are better observed by TEM, and the results are presented below.

The number average diameter (*d*_n_) was measured for all the blends with the three different types of added TiO_2_ nanoparticles: hydrophilic (hphi), hydrophobic (hpho), and hydrophobically modified (hphoM) TiO_2_. The values of the average sizes of the rPET droplets are shown in Table 3, and the average number diameters are plotted in Figure 6 as a function of composition. The average size dispersion of the particles (*D*) was also calculated according to equation 1 and reported in Table 3.

As can be appreciated in Figure 4 and Figure 5, PET particle size is reduced as TiO_2_ content increases. Additionally, Table 3 reports that particle size dispersion becomes narrower as TiO_2_ content increases. The hphoM TiO_2_ induces the highest reduction in rPET droplet size (see Figure 4), having a final particle size of about 2.7 µm with 12% TiO_2_, in comparison with the original PBNANO with 2% TiO_2_ with an average number diameter of 4.99 µm. In the case of hphi and hpho nanoparticles, the reduction in droplet size seems to saturate beyond 7% TiO_2_ (see Figure 4, Figure 5 and Figure 6). So, the chemical modification of the nanoparticles is promoting the creation of a physical barrier which avoids coalescence and reduces both particle size and size dispersity.

The functionalization of the nanoparticles could, in principle, also affect the interfacial adhesion. However, for the PBNANOs prepared here, the adhesion between phases does not seem to be significantly modified, as in all samples, characteristic holes can be observed (See Figure 4 and Figure 5) that are formed during cryogenic fracture interfacial failure (i.e., adhesive failure is seen instead of cohesive failure). Appendix A show the SEM micrographs with hphi and hpho 3%, 7%, and 12% TiO_2_ nanoparticles.

Although several theories have been proposed to explain the reduction of droplet size [14], the significant reduction in droplet diameters when 12% hydrophobically modified TiO_2_ nanoparticles are added, is due to the preferred location of such nanoparticles at the polymer–polymer interface, as evidenced by TEM below. Similar results have been previously reported for different PP binary blends. Fenouillot et al. studied the location of hydrophilic pyrogenic silica nanoparticles in PP/EVA blends. The particles migrated from the PP matrix to the EVA dispersed droplets. The authors proposed that collisions between silica particles and the dispersed EVA droplets were the predominant mechanism leading to the final nanoparticles location. They also performed hydrophobic surface treatments to the silica and then found that the hydrophobically modified nanoparticles were located in the PP matrix and at the interface [37]. Dubois et al. also demonstrated a reduction in the droplet size of the dispersed phase in 80 PP/20 PA and 80 PP/20 PC blends adding 5% of nanosilica [38].

TEM micrographs of the PBNANO-hphoM are shown in Figure 7, as the highest reduction of droplet size is seen by SEM with this type of TiO_2_. The nanoparticles are located inside the rPET droplets and the interface. As the content of TiO_2_ increases, the interface is progressively coated with nanoparticles. When 12% of hphoM TiO_2_ is used, the interface is almost completely covered by nanoparticles. Appendix A show the TEM micrographs with hydrophilic (hphi) and hydrophobic (hpho) 3%, 7%, and 12% TiO_2_ nanoparticles.

Figure 8 shows the difference in nanoparticle location using different types of TiO_2_ nanoparticles. In the case of hphi nanoparticles, they are located inside the rPET droplets because of their polar nature. The hpho nanoparticles are mostly located in the rPP matrix, although some of the nanoparticles can also be appreciated inside the rPET droplets and at the interface. More interestingly, in the case of the hphoM nanoparticles, the nanoparticles are preferentially located at the interface, as already mentioned above.

Several factors affect the final location of nanoparticles in PBNANOS, such as polymer-filler interactions, thermodynamic, and kinetic factors. Most of the times, it is difficult to determine which is the main factor. However, chemical modifications of the nanoparticles could explain its location. Such modifications promote interactions with one of the phases inducing thermodynamic stability. That is why the hydrophilic (hphi) nanoparticles are mostly located inside the rPET polar phase, whereas the hydrophobic (hpho) ones are preferably inside the rPP apolar matrix. However, the hydrophobically modified (hphoM) nanoparticles are located at the interface, probably because of kinetic factors. A balance between thermodynamics and kinetics is to be considered in the final location of nanoparticles [14].

It must be remembered that additional TiO_2_ nanoparticles were added by first preparing a masterbatch in rPP. Therefore, during extrusion, the TiO_2_ nanoparticles can migrate to the interface or to the PET droplets, depending on a balance between thermodynamics and kinetics. In the case of hydrophilic TiO_2_, the nanoparticles were able to migrate to the PET droplets (as shown in Figure 8a,d), as they were probably rejected from the apolar rPP matrix to the interface during extrusion and once at the interface, they were able to easily penetrate the hydrophilic PET droplets. On the other hand, commercial hydrophobic TiO_2_ nanoparticles are more affine to the rPP matrix and are expected to remain within the matrix because of the organic groups present on the surface of the particles. The existence of silanols, as well as methyl silane groups, on the surface of the hydrophobically modified TiO_2_, revealed by FTIR analysis, contributes to their location at the interface.

### 3.3. Thermal Characterization

#### 3.3.1. Non-Isothermal DSC Experiments

DSC cooling scans were performed after erasing thermal history, and then the samples were subsequently heated to record the heating DSC scans. The most relevant calorimetric parameters are listed in Appendix A, together with the DSC scans, which are reported in Appendix A. Experiments were performed at 20 °C/min. Enthalpies of crystallization (Δ*H_c_*) and melting (Δ*H_m_*) and the values of crystallinity degree (*X_c_*) were normalized by the weight fraction of the phase under consideration. The enthalpy of crystallization and fusion of 100% crystalline PP and PET were taken as 207 [39] and 140 J/g [40].

The results obtained by non-isothermal DSC corroborate with the immiscibility of the blends. The two phases, rPP and rPET, crystallize and melt separately, and no significant changes in their melting points were observed. In the case of the crystallization, rPP is not affected by TiO_2_ or PET addition, indicating that these PBNANOs components are not able to nucleate rPP. On the other hand, increases in peak crystallization temperature (of up to 10 °C) of the PET phase were obtained as a result of the nucleating action of TiO_2_ on the PET droplets.

#### 3.3.2. Isothermal DSC Experiments

Isothermal DSC experiments were performed to quantitatively measure the overall isothermal crystallization kinetics of both crystallizable phases, i.e., rPP and rPET. The inverse of the half-crystallization time is an experimental quantity proportional to the overall crystallization rate, in which both nucleation and growth contributions are included. Figure 9 shows the overall crystallization rate (expressed as 1/τ_50%_) as a function of temperature for neat rPP, and for the rPP matrix phase within all PBNANOs with 12% TiO_2_. In consonance with the non-isothermal results, the overall isothermal crystallization rate does not show any significant difference between neat rPP or the rPP phase within the PBNANOs. We can, therefore, conclude that neither the TiO_2_ nanoparticles nor the PET droplets can nucleate rPP.

The isothermal crystallization of the dispersed PET droplets within the PBNANOs was also studied. Figure 10 shows the results obtained together with representative TEM micrographs of the PET droplets. Both neat rPET and the rPET dispersed droplets within the PBNANOs with 2% of titanium dioxide crystallize at the same rate. This 2% TiO_2_ which was already included in rPET is not capable of nucleating PET droplets. According to TEM images, this 2% TiO_2_ is mostly located in the rPP phase, so the results are consistent with the lack of nucleation of the rPET droplets.

On the other hand, the nucleation effect on the rPET phase when additional TiO_2_ nanoparticles are added is evident. The overall crystallization rate increases with TiO_2_ content, and the magnitude of the effect is related to the preferential location of the nanoparticles. Hydrophilic TiO_2_ nanoparticles are mostly located inside the rPET droplets, and this causes a nucleation effect on rPET that accelerates its overall crystallization rate, as can be clearly seen in Figure 10. The maximum overall crystallization rate among all rPET droplets at low crystallization temperatures is obtained for the PBNANO containing 12% hydrophilic TiO_2_, as expected from the nanoparticles privileged location, i.e., well-dispersed inside the rPET droplets, thereby maximizing possible nucleation effects.

The rPET droplets within PBNANOs with 12% hydrophobic and hydrophobically modified TiO_2_ can also crystallize faster than neat rPET. Nevertheless, no large differences can be observed between them, as most of the nanoparticles in both cases stay in the rPP matrix or at the interface. For nucleation to occur, contact between the TiO_2_ nanoparticles and the rPET droplets is necessary. A higher overall crystallization rate is therefore correlated with the number of TiO_2_ nanoparticles in contact with rPET droplets. The results of Figure 10 are remarkably consistent with the morphology and location of the nanoparticles within the PBNANOs employed here.

Figure 11 shows TEM micrographs of the PBNANOs containing 12% hydrophilic (a,b), hydrophobic (c,d), and modified hydrophobic (e,f) TiO_2_. These micrographs were obtained after the sample was stained using a ruthenium tetroxide (RuO_4_) solution. The heavy RuO_4_ atoms can penetrate and preferentially stain the amorphous regions of the sample, while the denser crystalline regions will remain almost unstained. As a consequence, the lamellae inside the spherulites or axialites can be observed as the contrast between amorphous and crystalline regions of the sample make the crystalline lamellae easily visible. Interlamellar amorphous regions are seen as dark bands (which absorb more electrons) while the crystalline lamellae are preferentially white (as they are more transparent to the electron beam since they only contain the original atoms present in the polymer crystals, which are light atoms, C, H, and O). Dark regions at the interfaces between matrix and droplets are produced by interfacial staining and also by the presence of TiO_2_ nanoparticles at the interface. However, as these micrographs were obtained to highlight the lamellar morphology, the focus of the microscope was centered at the lamellae and not at the nanoparticles (so the nanoparticles are out of focus), as the TiO_2_ nanoparticles were already identified in unstained TEM observations (Figure 7 and Figure 8). Other dark spots in some of the images are the result of staining artefacts [42]. Figure 11 shows that crystalline lamellae and interlamellar amorphous regions can be clearly appreciated inside rPET droplets.

Figure 11a,b correspond to a PBNANO droplet (i.e., rPET) with 12% hydrophilic TiO_2_ (hphi). In this case, TiO_2_ nanoparticles are randomly dispersed within the rPET droplets (which cannot be seen in this micrograph but previously shown in Figure 8a). As a consequence, many edge-on lamellae are clearly visible inside the droplet with a random orientation within the droplet, as they have been nucleated inside the volume of the droplet by the TiO_2_ nanoparticles (see Figure 11b for a close-up showing in-volume nucleation of the lamellae), as in this case, the nanoparticles do not preferentially sit at the interface with rPP matrix.

On the contrary, Figure 11c shows a PBNANO droplet with 12% hydrophobic TiO_2_ (hpho). In this case, unstained TEM images showed that most of the nanoparticles concentrate inside the rPP matrix and at the interface between the phases (Figure 8b). Figure 11c shows some of the rPP lamellar texture within the matrix of the PBNANO, at the top of the micrograph, in the zone signaled by a white rectangle. In fact, a closer examination shows some cross-hatched lamellar morphology, which is typical of isotactic polypropylene [43]. We can also observe in Figure 11c the lamellar morphology of rPET (i.e., inside the droplet). In this case, a clear orientation can be observed in some lamellar stacks signaled by the bottom white rectangle. The lamellae are pointing towards the interface, where they were probably nucleated by TiO_2_ nanoparticles sitting at the interface. A close-up is shown in Figure 11d, with a higher magnification micrograph, presenting the parallel orientation of the lamellae pointing toward the particle interface.

Finally, Figure 11e corresponds to the PBNANO with 12% hydrophobically modified TiO_2_, where the nanoparticles are accumulated preferentially at the interface (Figure 8c). The morphology resembles that of Figure 11c. We have identified, by signaling with white rectangles, areas where oriented lamellar stacks seem to have nucleated at the interface and are oriented perpendicular to it (see Figure 11e,f).

The overall crystallization rate behavior can be usually described in the primary crystallization stage (i.e., the free growth stage, before spherulite impingement) with the well-known Avrami equation, which can be expressed, according to Lorenzo et al. as [44]:(2)$$1-V_{c}(t-t_{0})=\exp(-kt{\left(t-t_{0}\right)}^{n})$$
where *t* is the measurement time, *t*_0_ is the incubation time, *V_c_* is the relative volumetric transformed fraction, *n* is the Avrami index, and *k* is the overall crystallization rate constant. The fits to the Avrami equation were performed with the free Origin plug-in developed by Lorenzo et al. [32].

Figure 12 shows one example of the good agreement between the Avrami model and the experimental data obtained for neat rPP. Figure 12a compares the experimental and the predicted DSC isothermal scan. In this particular example, the agreement between the experiment and the Avrami theory goes beyond the peak value, i.e., beyond the primary crystallization regime. The predicted half crystallization time is practically the same as that experimentally measured, and the fitting of the theory is very good up to about 80% conversion (see Figure 12a,b). Figure 12c shows the typical Avrami plot for a limited conversion range. The Avrami equation can perfectly describe the overall crystallization kinetics within the first part of the primary crystallization range (with a correlation coefficient of 1.0000), during free superstructural growth, in a conversion range up to 20%.

Figure 13 shows the results obtained from the application of the Avrami equation for the isothermal crystallization of rPP and rPET phases within PBNANOs. The Avrami index value oscillates from 2.5 to 3.4 for rPP. This range can be approximated to n = 3, which is a value that suggests that rPP forms instantaneously nucleated spherulites [32]. The Avrami index obtained in the case of rPET is in the range 2.25 to 3.75. Below n = 2.5, the formation of axialites is predominant, in some of the PBNANO samples with hydrophobically modified TiO_2_ nanoparticles. However, most of the other PBNANOs have Avrami indexes close to 3, once again indicating the presence of instantaneously nucleated spherulites.

### 3.4. Mechanical Behavior

Figure 14a shows the representative engineering stress–strain curves of PBNANOs, while in Figure 14b the appearance of the material after being tested is presented. PBNANO-0 is the sample without additional TiO_2_, which contains 2% TiO_2_, as the rPET employed in this work already had 2% TiO_2_. So, PBNANO-0 is the 80/20 blend of rPP and rPET without any additional TiO_2_, which was first extruded and then compression molded into sheets, from which tensile testing bars were obtained by stamping.

Two of the samples shown in Figure 14 exhibit a ductile behavior with cold drawing. This means that after reaching a local maximum in the engineering tension, associated with the yield point, there is a decrease in tension. After the maximum in the stress–strain curve, at higher strains, the tension stabilizes. This phenomenon can be associated with the formation of a neck and its subsequent stable propagation. In the case of the PBNANO-0 and PBNANO-hphoM, although a yield point is reached, the neck stabilization stage does not occur.

Figure 14b shows that only two of the PBNANOs (without and with hydrophilic TiO_2_) exhibit neck propagation accompanied by stress whitening. In the case of systems with hydrophobic TiO_2_ (without and with modification), the necking and the degree of stress whitening is negligible (hpho) and even non-existent (hphoM). In polymeric systems with dispersed phases with different stiffness than the matrix containing it, stress whitening is usually associated with a de-cohesion/cavitation process of the dispersed particles.

Figure 15 presents SEM micrographs of the fracture zone of the tensile specimens of the different materials. The observation by SEM of the fracture surfaces after the test (Figure 15) reveals that for both PBNANO-0 (Figure 15a) and the PBNANO-hphi (Figure 15d), the ratio of de-cohesion and cavitation is high (not for PBNANO-hpho and PBNANO-hphoM in Figure 15b,c, respectively), and even the rPP matrix has a high degree of tearing, which decreases with increase in the added quantities of additional TiO_2_. It should be noted, that the dispersed phase (rPET-O) shows no significant deformation, indicating that this phase has not reached its yield process, acting as a stiff inclusion in the blend. All these situations indicate that precisely this process of cavitation is helping to alleviate the degree of local triaxiality, promoting the yielding of the matrix. On the contrary, in the case of the PBNANO-hpho and PBNANO-hphoM, the degree of de-cohesion of the dispersed phase is appreciably lower, and even with evidence of extensive tearing of the matrix.

Figure 16 shows the tensile mechanical parameters obtained. Also, the predictions of the parameters have been represented considering the additive mixing law (AML) at its upper limit (i.e., assuming that the adhesion is good, and therefore both phases are deformed equally) and its lower limit (i.e., both phases bear the same tension, but as a consequence of a moderate adhesion and/or disposition of the dispersed phase, each one is deformed independently).

Considering the elastic range of the material, through the elastic modulus, *E*, (Figure 16a), it can be seen that PBNANO-0 complies with the AML prediction at its upper limit, suggesting that the degree of adhesion achieved is sufficient to offer an effective reinforcement in the elastic range of the mechanical behavior of the material. When adding TiO_2_, independently of its hydrophobic nature, there is a positive deviation in the trend, registering an increase, with respect to PBNANO-0, of up to 13% in the presence of hydrophobic TiO_2_ and of 6% in the other cases. This increase could be expected given the addition of a stiffer phase (TiO_2_) at a relatively high content.

Regarding the yield stress (σ_y_), associated with the onset of plastic behavior (Figure 16b), a negative deviation of the AML at its lower limit is observed for all cases, and without a significant dependence of the type of TiO_2_ added. This observation confirms the triaxiality relief effect exerted by the de-cohesion of the dispersed phase, promoting the yield of the matrix (rPP). When considering the deformation registered at this point for each material (ε_y_), it is observed how the PBNANOs with TiO_2_ of hydrophobic character show a marked negative deviation of AML in its lower limit, while the PBNANO and the PBNANO-hphi practically comply with this prediction.

It is well-known that in multiphasic systems, the de-cohesion between phases, during a tensile test, involves loss of stress transfer, so that the matrix begins to bear the entire load from that moment onward. So the nature and strength of the interface that is generated between the phases acquire an important role.

As discussed in previous sections, the preferential location of TiO_2_ in the rPP-rPET-O interface is dictated by the degree of hydrophobicity that the particle presents. In the case of hydrophilic TiO_2_, the nanoparticles are located preferentially within the PET phase, while those of a hydrophobic character in the outer zone of the PET phase (Figure 8). Figure 17 shows micrographs of the fracture zone of tensile specimens where this situation is once again evident. It can be seen that in the case of PBNANO-hphoM (Figure 17a), the particle of rPET-O is coated by TiO_2_ on its outer face, and even after cavitation, a “bed” of TiO_2_ particles prevails (see the yellow circle in Figure 17a). A similar situation has been observed for the PBNANO-hpho (not shown). In the case of the PBNANO-hphi, the rPET-O seems to have some nanoparticles on the surface, but after de-cohesion, no nanoparticles are left on the “bed.”

The results presented above show that for the case of PBNANO-hpho and PBNANO-hphoM, the interface acts as a third phase with very low shear resistance. Perhaps the surface treatment applied promotes a lubrication effect between particles, facilitating the de-cohesion at relatively low global deformation levels, which limits the stress transfer between the phases. The latter, combined with the emulsifying effect exerted on the rPET-O behavior, could cause this phase to act as a stress concentrator, accelerating the collapse and rupture of the material.

In the case of the PBNANO-hphi, all seems to indicate that this “third phase” of lower resistance (with lubricating effect) does not exist (as most nanoparticles are inside the PET droplets), so that the relative adhesion is higher through a merely mechanical anchoring mechanism between rough surfaces, which would allow a slight improvement in the stress transfer between phases. This effect is evidenced at the moment of fracture (Figure 16d). As it can be observed, the rupture deformation (ε_b_) of the PBNANO-0 and PBNANO-hphi systems are within the prediction of the lower limit of the AML, while the systems with hydrophobic load are located far below of this limit, being much smaller in the material with additional hydrophobic treatment. The tensile properties for all materials are also reported in Appendix A.

## 4. Conclusions

The degree of hydrophilicity of the three types of TiO_2_ employed decreases in the order hphi > hpho > hphoM. A general reduction of rPET droplet size and droplet size dispersion occurs with the addition of TiO_2_ nanoparticles. A hydrophilic/hydrophobic balance controls the nanoparticles location. TEM shows that hphi TiO_2_ preferentially locates inside the PET droplets and hpho is found at the interface and within the PP matrix. HphoM TiO_2_ also locates within the PP matrix and at the interface, but at large loadings (12%) it can completely cover the surfaces of the droplets forming a physical barrier that avoids coalescence during blending, leading to the formation of smaller droplets.

While TiO_2_ does not nucleate PP and therefore does not significantly influence its crystallization rate, the added TiO_2_ nanoparticles do nucleate the PET droplets, and the nucleation effect increases with particle loading. A good correlation is found between the crystallization rate of PET (determined by DSC) and nanoparticles location, where hphi TiO_2_ induces the highest PET crystallization rate, as it is preferentially located inside the PET droplets. PET lamellar morphology (revealed by TEM) is also dependent on particle location.

In the case of the PBNANO with hphi TiO_2_, edge-on lamellae with random orientation were visualized by TEM inside rPET droplets, because TiO_2_ nanoparticles (that are preferentially located inside the droplet volume) randomly nucleate rPET droplet. For the case of the hydrophobic TiO_2_ nanoparticles, we were able to see PET lamellae nucleating at the interface of the droplets, where hydrophobic nanoparticles preferentially locate.

The surface treatment of TiO_2_ particles that confers the hydrophobic character generates a low resistance interface that promotes a greater de-cohesion of the dispersed phase. This causes a decrease in the tensile strength and accelerates the collapse and subsequent rupture of the system. However, it shows a clear efficiency as mechanical reinforcement in the elastic range of the mechanical behavior of the system.

As PET and other polyolefins such as PP and PE are commonly used in packaging, very high amounts of these materials have to be recycled in order to decrease waste accumulation. Because of the presence of titanium dioxide nanoparticles, recycling is more complicated and that is why it is interesting to find new applications for these materials, as they could be upcycled for instance into automotive parts.

## Figures and Tables

**Figure 1 polymers-11-01692-f001:**
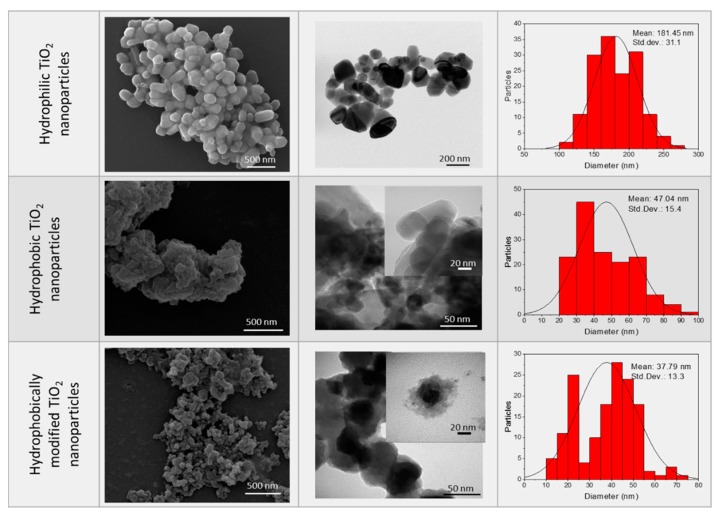
Scanning electron microscopy (SEM) and transmission electron microscopy (TEM) images and size histograms of the three different types of TiO_2_ nanoparticles. Hydrophobic samples were negatively stained with phosphotungstic acid for TEM analysis.

**Figure 2 polymers-11-01692-f002:**
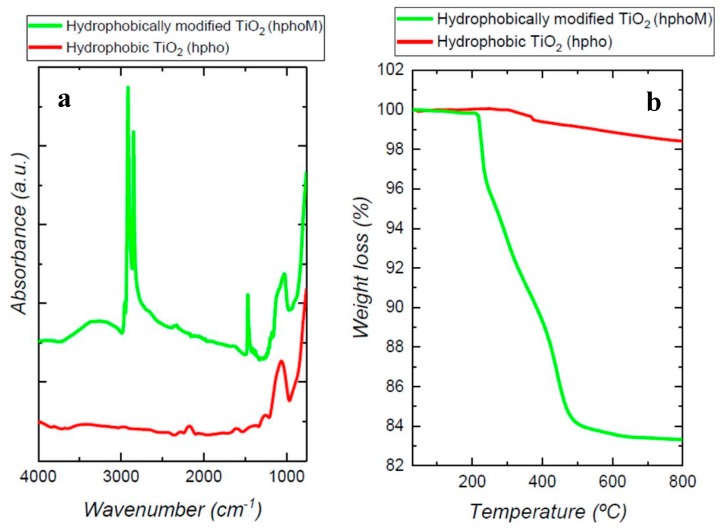
(**a**) Fourier-transform infrared (FTIR) spectra and (**b**) thermogravimetric analysis TGA curves of commercial hydrophobic (hpho) and TMOS modified nanoparticles (hphoM).

**Figure 3 polymers-11-01692-f003:**
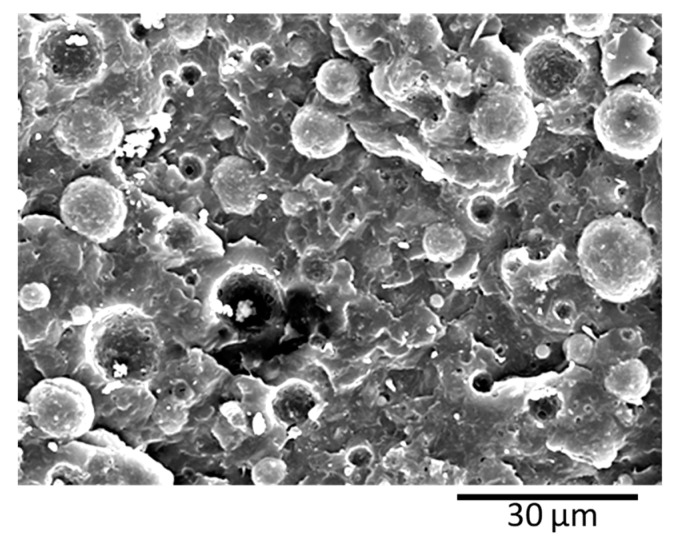
SEM image of PBNANO-2% TiO_2_.

**Figure 4 polymers-11-01692-f004:**
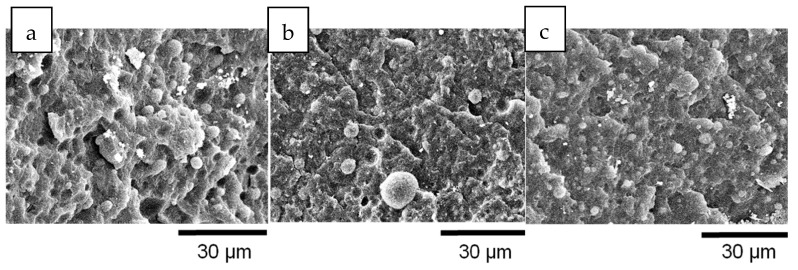
SEM images of PBNANO-12% containing: (**a**) hydrophilic (hphi), (**b**) hydrophobic (hpho), and (**c**) hydrophobically modified (hphoM) TiO_2_.

**Figure 5 polymers-11-01692-f005:**
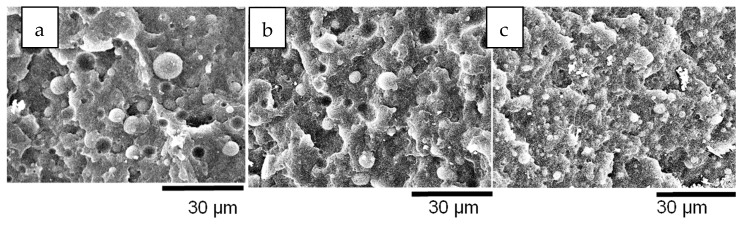
SEM images of PBNANO-hphoM with (**a**) 3% (**b**) 7%, and (**c**) 12% TiO_2_.

**Figure 6 polymers-11-01692-f006:**
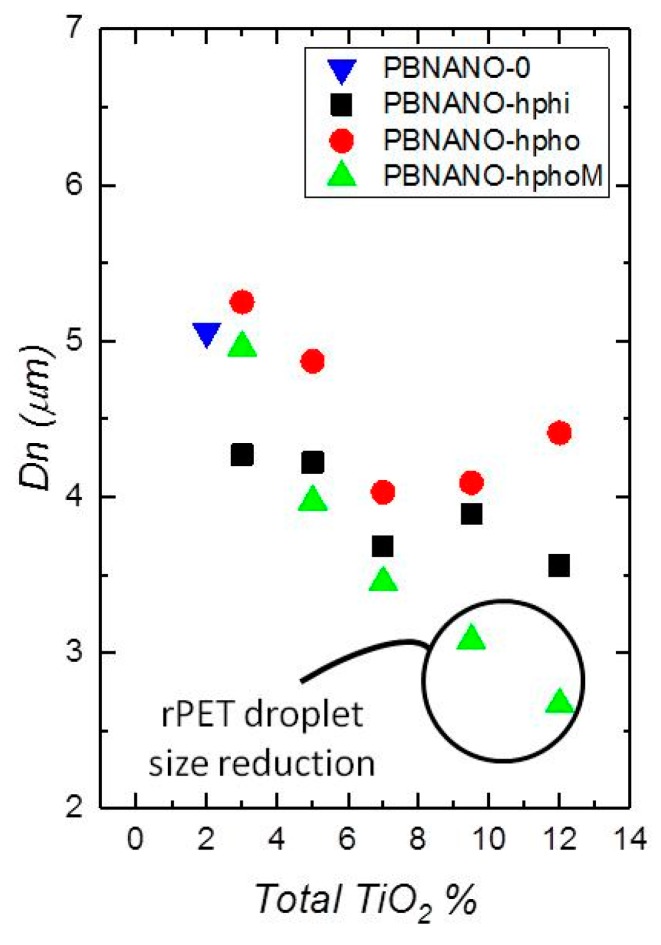
Number average diameter *(d_n_)* plotted as a function of total TiO_2_ content for the PBNANOs prepared in this work with the three types of TiO_2_ indicated in the legend. PBNANO-0 is the sample without additional TiO_2_, which contains 2% TiO_2_, as the rPET employed in this work already had 2% TiO_2_. So, PBNANO-0 is the 80/20 blend of rPP and rPET without any additional TiO_2_.

**Figure 7 polymers-11-01692-f007:**
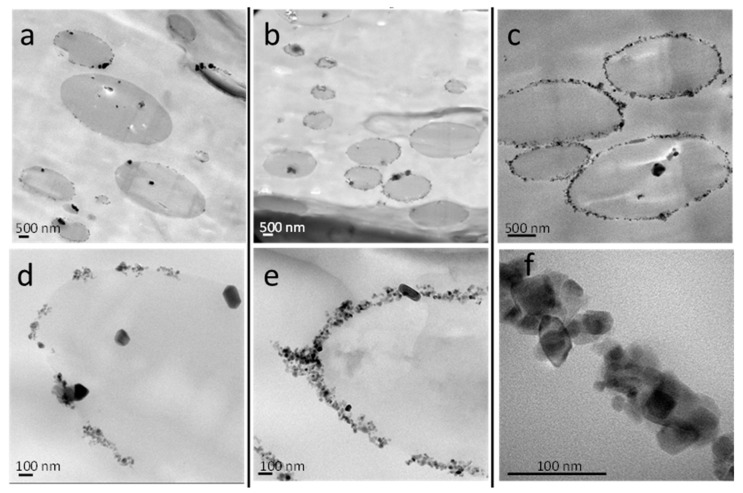
TEM images of PBNANO-hphoM with: (**a**,**d**) 3% (**b**,**e**) 7%, and (**c**,**f**) 12% TiO_2_.

**Figure 8 polymers-11-01692-f008:**
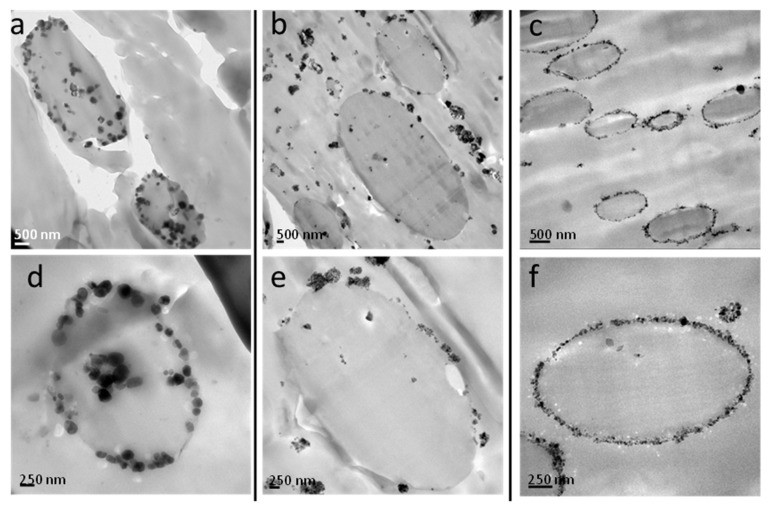
TEM images of PBNANO-12% containing: (**a**,**d**) hydrophilic (hphi), (**b**,**e**) hydrophobic (hpho), and (**c**,**f**) hydrophobically modified (hphoM) TiO_2_.

**Figure 9 polymers-11-01692-f009:**
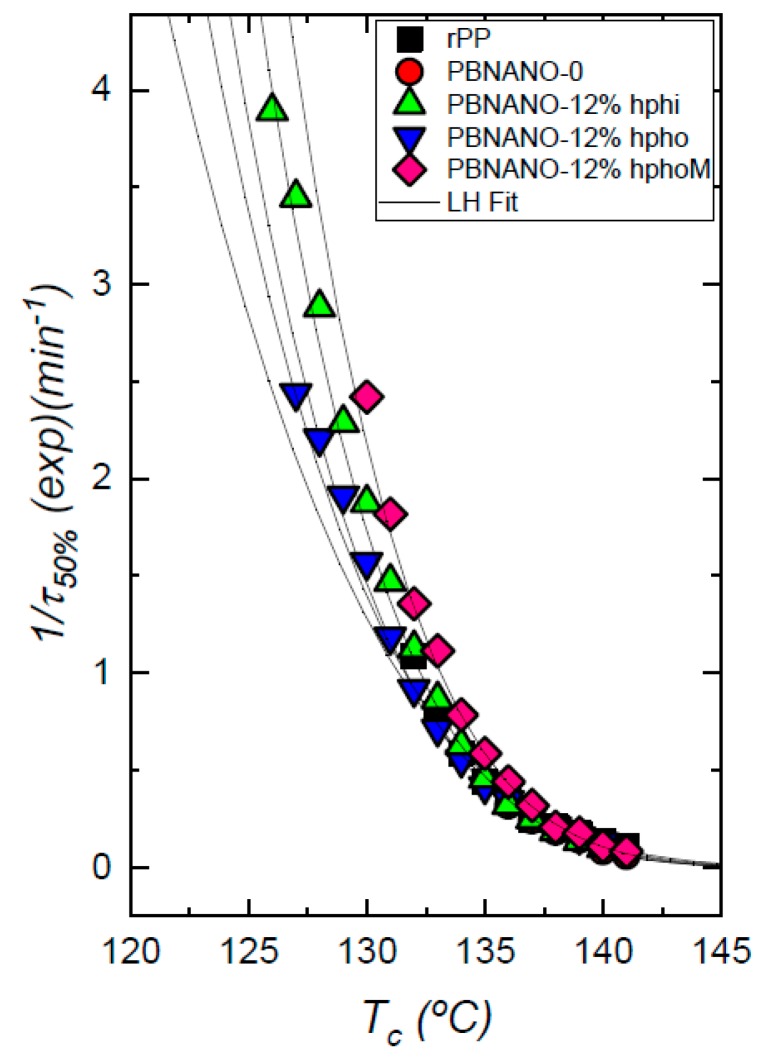
Overall crystallization rate of rPP and the rPP phase within the indicated PBNANOs as a function of crystallization temperature. Solid lines correspond to mathematical fits to the Lauritzen and Hoffman theory [41].

**Figure 10 polymers-11-01692-f010:**
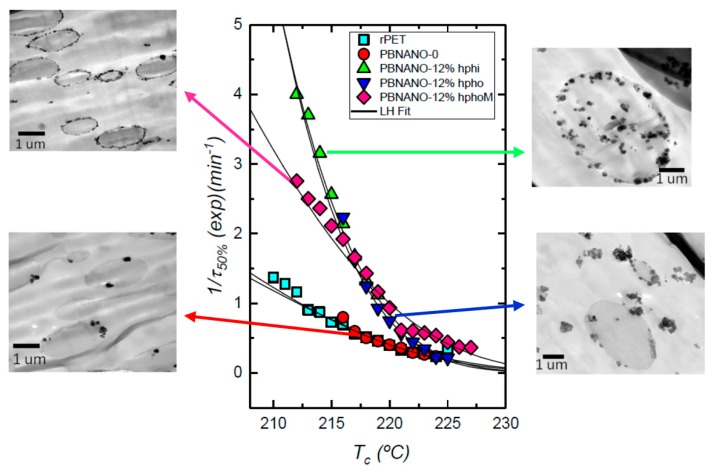
Overall crystallization rate of rPET and the rPET phase within the indicated PBNANOs as a function of crystallization temperature. Solid lines correspond to mathematical fits to the Lauritzen and Hoffman theory [41].

**Figure 11 polymers-11-01692-f011:**
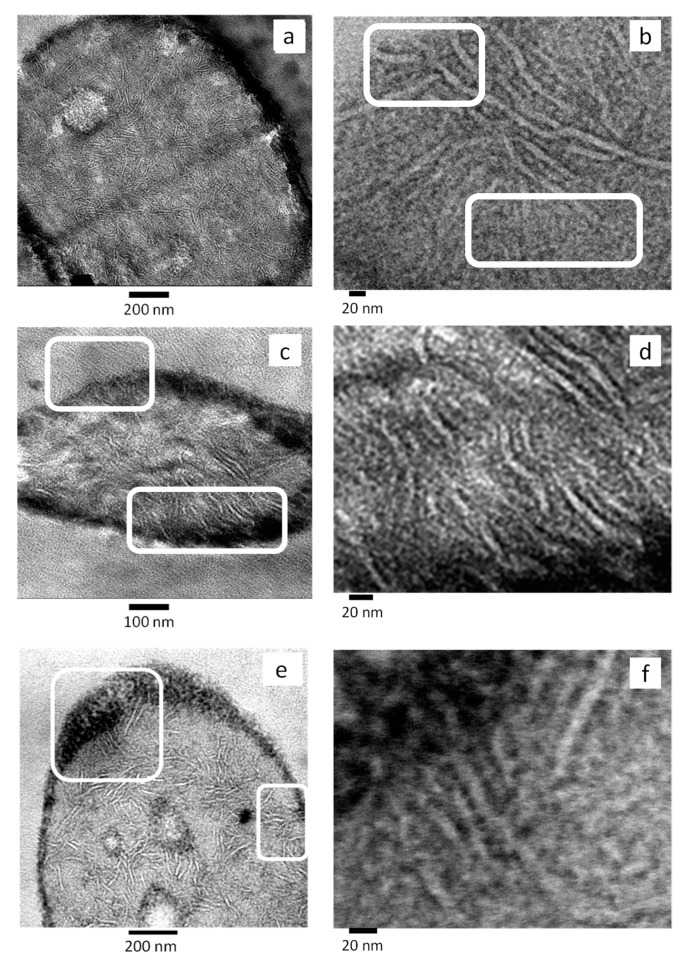
TEM images of 80rPP/20rPET/12% TiO_2_ PBNANOs containing: (**a**,**b**) hydrophilic TiO_2_ (hphi); (**c**,**d**) hydrophobic TiO_2_ (hpho); (**e**) and (**f**) hydrophobically modified TiO_2_ (hphoM).

**Figure 12 polymers-11-01692-f012:**
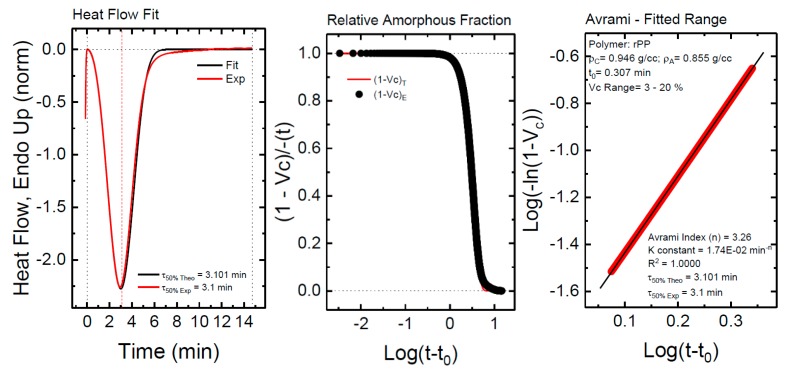
(**a**–**c**) Comparison between experimental data and the fits to the Avrami equation using the Origin plug-in developed by Lorenzo et al. [32].

**Figure 13 polymers-11-01692-f013:**
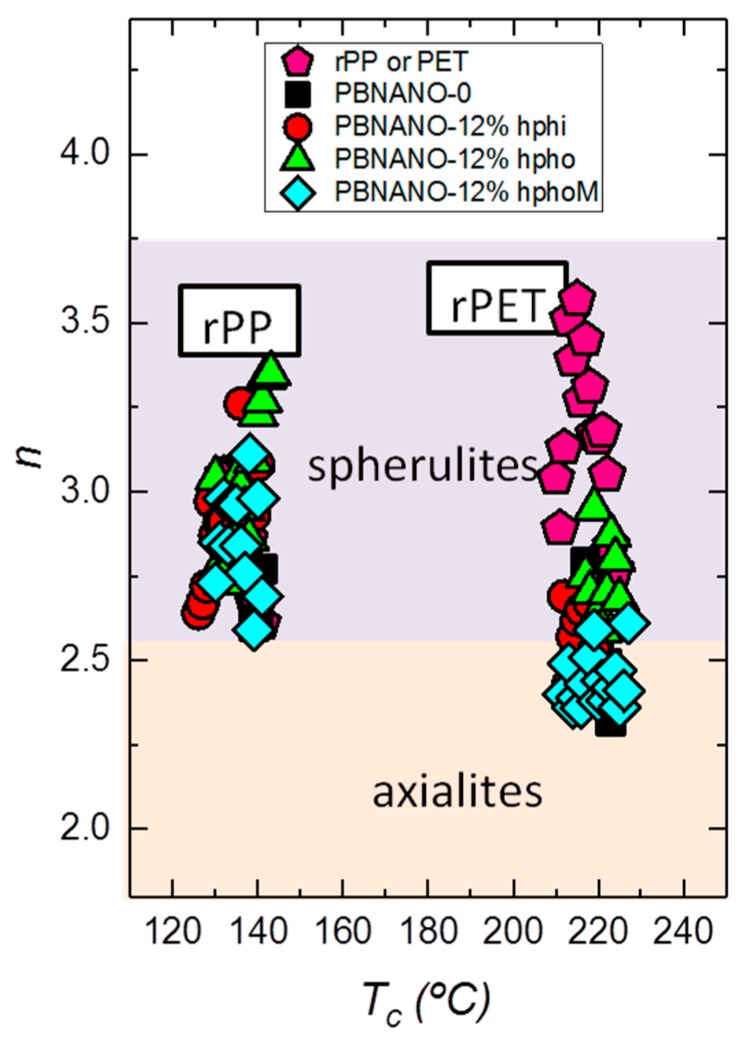
Avrami index (n) as a function of crystallization temperature for the indicated samples.

**Figure 14 polymers-11-01692-f014:**
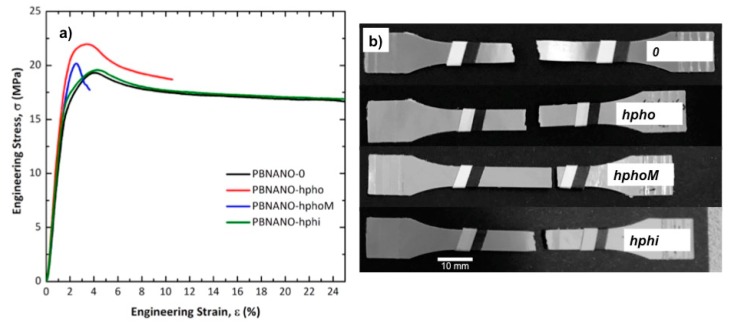
Representative engineering stress–strain curves of PBNANOs tested (**a**) and appearance of the deformation process zone after tensile tests (**b**).

**Figure 15 polymers-11-01692-f015:**
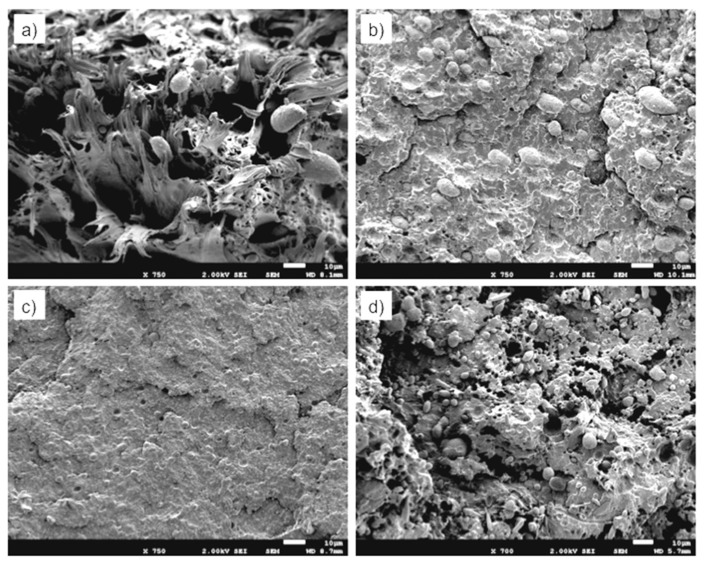
SEM micrographs of the fracture process zone after tensile tests of PBNANO-0 (**a**) PBNANO-hpho (**b**), PBNANO-hphoM (**c**) PBNANO-hphi (**d**).

**Figure 16 polymers-11-01692-f016:**
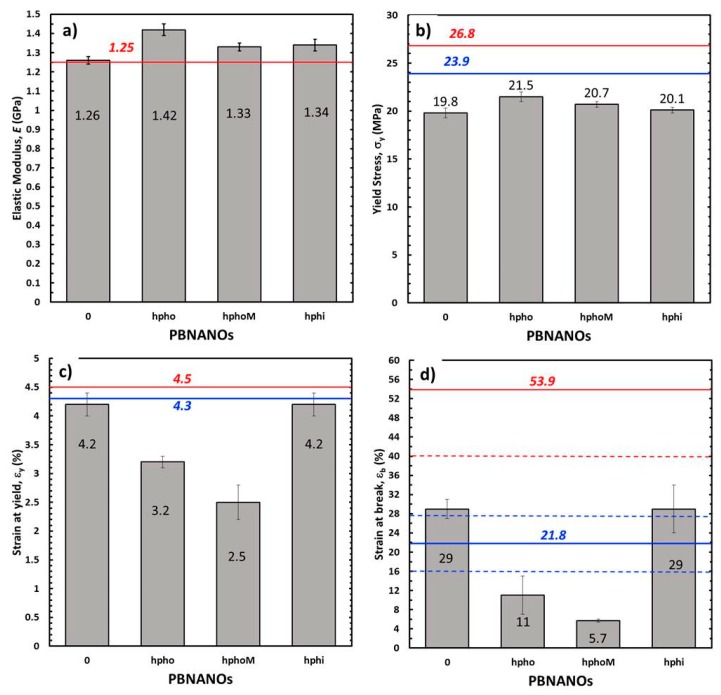
Mechanical tensile parameters obtained for the materials under study. (**a**) Elastic modulus (E); (**b**) yielding stress (σ_y_); (**c**) deformation at yield (ε_y_) and (**d**) elongation at break (ε_b_). Horizontal solid lines represent the upper (red) and the lower (blue) limits of the additive mixing law. The dashed lines in (**d**), represent their respective error bands.

**Figure 17 polymers-11-01692-f017:**
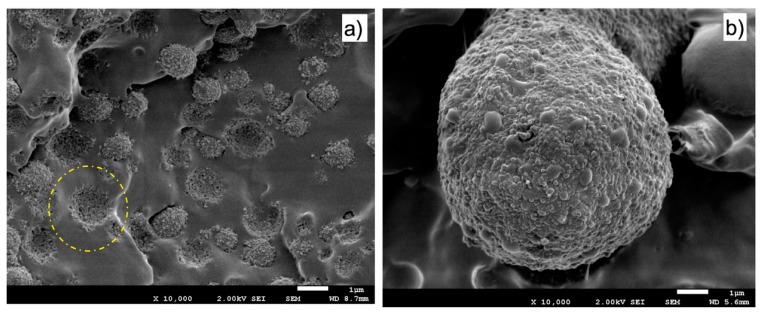
SEM micrographs from fracture surface of: PBNANO-hphoM (**a**) and PBNANO-hphi (**b**). The yellow circle highlights the TiO_2_ nanoparticles that are left attached to the rPP matrix (“bed”) after extraction of a PET droplet.

**Table 1 polymers-11-01692-t001:** Compositions of the prepared polymer blend nanocomposites (PBNANOs) in *w*/*w*.

Samples	Added TiO_2_	Total Amount of TiO_2_
80rPP/20rPET/TiO_2_	0	2
80rPP/20rPET/TiO_2_	1	3
80rPP/20rPET/TiO_2_	3	5
80rPP/20rPET/TiO_2_	5	7
80rPP/20rPET/TiO_2_	7.5	9.5
80rPP/20rPET/TiO_2_	10	12

**Table 2 polymers-11-01692-t002:** Hydrodynamic size and ζ-potential of TiO_2_ nanoparticles.

Nanoparticle	Hydrodynamic Diameter (nm)	ζ-Potential (mV)	Contact Angle (°)
Hydrophilic (hphi) ^1^	170 ± 116	7.2 ± 1.1	36 ± 4
Un-modified ^2^	73 ± 21	10.1 ± 0.4	0
Hydrophobically modified (hphoM) ^3^	104 ± 54	−31.1 ± 0.6	144 ± 5
Hydrophobic (hpho) ^4^	541 ± 13	−9.1 ± 1.9	106 ± 1

^1^ Hphi was provided by “io-li-tec” nanomaterials with rutile crystalline form. ^2^ Supplied by AEROXIDE TiO_2_ P25, from Evonik Industries. ^3^ AEROXIDE TiO_2_ P25, from *Evonik Industries* modified by a sol-gel reaction with octadecyltrimethoxysilane to turn it hydrophobic (hphoM). ^4^ Hydrophobic titanium dioxide (hpho) coated with silicone oil from US Research Nanomaterials, Inc., with rutile crystalline form.

**Table 3 polymers-11-01692-t003:** Number-average diameter *(d_n_)* and particle size distribution *(D)* of 80rPP/20rPET/TiO_2_ nanocomposite blends.

Total TiO_2%_	Hydrophilic (hphi)		Hydrophobic (hpho)		Hydrophobically Modified (hphoM)	
	*Dn* (µm)	*D*	*Dn* (µm)	*D*	*Dn* (µm)	*D*
**2**			4.99	2.63		
**3**	4.27	1.21	5.25	1.55	4.96	1.39
**5**	4.22	1.49	4.87	1.64	3.97	1.27
**7**	3.68	1.17	4.03	1.74	3.45	1.22
**9.5**	3.89	1.36	4.09	1.19	3.08	0.69
**12**	3.56	0.93	4.49	1.31	2.67	0.61

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
