# Peer review of "The Effect of Titanium Dioxide Surface Modification on the Dispersion, Morphology, and Mechanical Properties of Recycled PP/PET/TiO2 PBNANOs"

_polymers, 2019, doi:10.3390/polym11101692_

Round 1

Reviewer 1 Report

Reviewer’s comments on the manuscript

The effect of titanium dioxide surface modification on the dispersion, morphology and mechanical properties of recycled PP/PET/TiO2 PBNANOs

This interesting paper deals with a blend constituted of a recycled PP matrix (80%), a recycled PET dispersed phase (20%) and TiO2 particles with different levels of hydrophilicity. This difference is shown to play a key role in the location of nanofillers, which governs morphology, crystallization kinetics and mechanical properties of polymer blend nanocomposites. Several fractions of nanofillers were studied.

Remarks:

Recent works concerning polyethylene (80%)/ polyamide (20%) blends filled with clay nanoparticles (organically modified montmorillonite) could enrich this paper, not only about the location and migration of nanoparticles within a polymer blend but also about the mechanical properties of systems with an interphase.

The “materials and methods” part should be separated into two parts:

- a material part (2.1), consisting of three sub-paragraphs (2.1.1 materials, 2.1.2 TiO2 nanoparticles modification and 2.1.3 PBNANOs preparation)

- a characterization part, in which the three paragraphs (2.7, 2.8 and 2.9) devoted to DSC characterization, should only be one.

Why do the authors give the expressions of the volume average diameter and the particle size polydispersity since they do not use them later in the results part?

Correct the difference in values: 542 ± 13 nm in the text (l. 236) against 541 ± 13 nm in Table 2. One particle weight loss curve (Fig; 2b) is above 100% between room temperature and 200°C. Do the authors have an explanation?

The blend morphology characterization part should first present SEM micrographs (Figs 3, 5 and 6), then the Table 3 with the number average diameters of all the materials studied, including that of the reference material containing 2% of particles, and finally the evolution of the number average diameter (Fig. 4). The scale of Figure 3 is not expressed in mm. The average diameter of reference material containing 2% of particles is reported equal to 4.27 mm (l. 296), but it is about 5mm in Fig 4: this point needs to be clarified.

From TEM micrographs, is it possible to determine a coverage coefficient of the interface, which has already been used in literature? This parameter plays a key role in the viscoelastic properties of the interphase, and consequently in the end-use properties of the materials.

The abscissa and the legends of Figs 9 and 10 are hidden and incomplete respectively.

The caption of Fig. 11 and its presentation in text (lines 438-439) are incorrect.

A line break is necessary at the line 475 (Finally, Figure 11e, corresponds ...). In addition, the comma after Figure 11e must be removed.

Why was the nanoparticle fraction chosen equal to 10% for mechanical tests? It would have been interesting to test materials without interphase, materials with a partial interphase and materials with a fully developed interphase. I think that Figures 15b and 15c are not cited in the text.

Considering the above-points, I do not recommend acceptance of the paper. Major revisions, in terms of organization of the text, scientific rigor concerning some results and quality of figures are expected. The interest of the nanoparticle fraction parameter must be specified.

Reviewer 2 Report

Dear Editor

I accurately reviewed the article

Manuscript Number: polymers-569100-peerreview-v1

Title: The effect of titanium dioxide surface modification on the dispersion, morphology and mechanical properties of recycled PP/PET/TiO2 PBNANOs

submitted to Polymers.

The authors investigated polymer blend nanocomposites (PBNANOs). They prepared composites by adding hydrophilic (hphi), hydrophobic (hpho), and hydrophobically modified (hphoM) titanium dioxide (TiO2) nanoparticles to 80rPP/20rPET recycled blends. Contact angle studies show that the degree of hydrophilicity of TiO2 decreases in the order hphi>hpho>hphoM. A reduction of rPET droplet size occurs with the addition of TiO2 nanoparticles. TEM shows that hphi TiO2 preferentially locates inside the PET droplets and hpho at both interface and PP matrix. HphoM also locates in the PP matrix and at the interface, but large loadings (12%) can completely cover the surfaces of the droplets forming a physical barrier that avoids coalescence leading to the formation of smaller droplets. A correlation is found between the crystallization rate of PET (determined by DSC) and nanoparticles location, where hphi TiO2 induces the highest PET crystallization rate. PET lamellar morphology (revealed by TEM) is also dependent on particle location. The mechanical behavior improves in the elastic regime with TiO2 addition, but the plastic deformation of the material is limited and strongly depends on the type of TiO2 employed. 

The topic is of great interest and combines physical and chemical properties of composite material with relevant application aspects.

Despite this, the authors should resolve some critical issues.

Introduction

The Introduction is lacking on the oxides composite nanomaterials, such as titania, preparation and applications. It would be useful for readers to have a panorama on these topics with some references, such as, just for example:

Composite Titanium Dioxide Nanomaterials; Chem. Rev.2014114199853-9889 Electrochemical and photoelectrochemical properties of screen-printed nickel oxide thin films obtained from precursor pastes with different compositions; Journal of The Electrochemical Society, 164 (2), (2017) H137-H147 Effect of titanium dioxide nanoparticles on mechanical properties of vinyl ester-based nanocomposites; Vol 49, Issue 19, 2015 Photoelectrochemical characterization of squaraine-sensitized nickel oxide cathodes deposited via screen-printing for p-type dye-sensitized solar cells; Applied Surface Science 356 (2015) 911-920 Porous Gig-Lox TiO2 Doped with N2 at Room Temperature for P-Type Response to Ethanol; Chemosensors 2019, 7(1), 12 Platinum nanoparticles on electrospun titania nanofibers as hydrogen sensing material working at room temperature; Nanoscale 6 (2014) 9177-9184 Preparation of Microporous Polypropylene/Titanium Dioxide Composite Membranes with Enhanced Electrolyte Uptake Capability via Melt Extruding and Stretching; Polymers 2017, 9(3), 110;

Experimental

More details are necessary about samples preparations and characterizations:

What is Suez?

Is it a company? Please provide some details

The work would be more complete comparing the data also with PET without titania

What is  io-li-tec nanomaterials?

Is it a company? Please provide some details

what is the purity grade of the reagents used?

Which instrument was used for DLS measurements?

what is the concentration used for DLS TEM and AFM measurements?

FT-IR?

specify TMOS

This information guarantees the repeatability of the experiments

Conclusions

The conclusions should be streamlined, highlighting the effect of the different nanotitania and its % on the final properties of the composites.

Furthermore, the applicative perspectives of this research should be indicated

Figures

Figure3, Figure 7, figure 9, figure 12 and figure 15 could be moved in Supporting materials.

Figure 9 and figure 10 are cut

Figure 16 needs improvement in resolution

English needs some improvements: some sentences are too long and there are some typos.

In conclusion, the article is suitable for publication in Polymers, but after major revisions.

best regards

Round 2

Reviewer 2 Report

Dear Editor,

I accurately reviewed the article

Title The effect of titanium dioxide surface modification on the dispersion, morphology and mechanical properties of recycled PP/PET/TiO2 PBNANOs

submitted to Polymers.

The authors have solved some problems but not all.

In my opinion the introduction remains poor and lacking.

The Introduction is lacking on the oxides composite nanomaterials, such as titania, preparation and applications. It would be useful for readers to have a panorama on these topics with some references, such as, just for example (naturally the authors can use other references, freely):

Composite Titanium Dioxide Nanomaterials; Chem. Rev.2014114199853-9889

Electrochemical and photoelectrochemical properties of screen-printed nickel oxide thin films obtained from precursor pastes with different compositions; Journal of The Electrochemical Society, 164 (2), (2017) H137-H147 

Effect of titanium dioxide nanoparticles on mechanical properties of vinyl ester-based nanocomposites; Vol 49,Issue 19, 2015 

Photoelectrochemical characterization of squaraine-sensitized nickel oxide cathodes deposited via screen-printing for p-type dye-sensitized solar cells; Applied Surface Science 356 (2015) 911-920 

Porous Gig-Lox TiO2 Doped with N2 at Room Temperature for P-Type Response to Ethanol; Chemosensors 2019, 7(1), 12

Platinum nanoparticles on electrospun titania nanofibers as hydrogen sensing material working at room temperature; Nanoscale 6 (2014) 9177-9184 

Preparation of Microporous Polypropylene/Titanium Dioxide Composite Membranes with Enhanced Electrolyte Uptake Capability via Melt Extruding and Stretching; Polymers 2017, 9(3),110

The figures are too many and weigh down the reading, some should be moved to the supporting material. Figure3, Figure 7, figure 9, figure 12 and figure 15 (or a part of these) could be moved in Supporting materials.

Figures 9 and 10 if they are not cut, they lack an axis

The characterizations should however have as a reference a material without titania and then compare those with different%.

If the authors fail to solve these problems, the work remains poor

In conclusion, the article will be published only after major revisions.

best regards

Author Response

The authors have solved some problems but not all.

WE HAVE THOROUGHLY REVISED THE MANUSCRIPT AND IN OUR OPINION, ALL RELEVANT PROBLEMS HAVE BEEN SOLVED.

In my opinion the introduction remains poor and lacking.

The Introduction is lacking on the oxides composite nanomaterials, such as titania, preparation and applications. It would be useful for readers to have a panorama on these topics with some references, such as, just for example (naturally the authors can use other references, freely):

Composite Titanium Dioxide Nanomaterials; Chem. Rev.2014114199853-9889

Electrochemical and photoelectrochemical properties of screen-printed nickel oxide thin films obtained from precursor pastes with different compositions; Journal of The Electrochemical Society, 164 (2), (2017) H137-H147 

Effect of titanium dioxide nanoparticles on mechanical properties of vinyl ester-based nanocomposites; Vol 49,Issue 19, 2015 

Photoelectrochemical characterization of squaraine-sensitized nickel oxide cathodes deposited via screen-printing for p-type dye-sensitized solar cells; Applied Surface Science 356 (2015) 911-920 

Porous Gig-Lox TiO2 Doped with N2 at Room Temperature for P-Type Response to Ethanol; Chemosensors 2019, 7(1), 12

Platinum nanoparticles on electrospun titania nanofibers as hydrogen sensing material working at room temperature; Nanoscale 6 (2014) 9177-9184 

Preparation of Microporous Polypropylene/Titanium Dioxide Composite Membranes with Enhanced Electrolyte Uptake Capability via Melt Extruding and Stretching; Polymers 2017, 9(3),110

WE RESPECTFULLY STRONGLY DISAGREE WITH THE REVIEWER. WE FEEL THESE REFERENCES ARE NOT RELEVANT FOR OUR PAPER AND WE ARE NOT GOING TO CITE THEM. THE REFEREE SEEMS TO HAVE A CONFLICT OF INTEREST BY HIS INSISTENCE ON ASKING US TO QUOTE THESE REFERENCES. 

The figures are too many and weigh down the reading, some should be moved to the supporting material. Figure3, Figure 7, figure 9, figure 12 and figure 15 (or a part of these) could be moved in Supporting materials.

ONCE AGAIN WE DISAGREE, WE FEEL ALL THESE FIGURES ARE RELEVANT FOR THE PAPER (AND REVIEWER 1 DID NOT ASK FOR THE FIGURES TO BE MOVED, SO WE ARE NOT ALONE IN THINKING THESE FIGURES ARE RELEVANT)

Figures 9 and 10 if they are not cut, they lack an axis

IN THE WORD FILE, THE FIGURE APPEAR PERFECT. THE PROBLEM MUST BE OF THE VERSION POLYMERS IS PREPARING IN PDF, WHICH HAS A POOR RESOLUTION

The characterizations should however have as a reference a material without titania and then compare those with different%.

WE ALREADY ANSWERED THIS QUESTION. OUR WORK DEALS WITH THE RECYCLING OF OPAQUE PET.

If the authors fail to solve these problems, the work remains poor

WE DISAGREE

In conclusion, the article will be published only after major revisions.

WE CONCLUDE THAT THE REFEREE HAS A CONFLICT OF INTEREST WITH US AND WE WOULD PREFER THAT OUR PAPER IS NOT HANDLED BY HIM/HER.

best regards